Navigating cross-cultural research:
methodological and ethical considerations.
*Proc. R. Soc. B* **287**: 20201245.

behaviour, evolution, cognition

cross-cultural research, ethics, evolutionary
anthropology, psychology

**Author for correspondence:**
Monique Borgerhoff Mulder
e-mail: mborgerhoffmulder@ucdavis.edu

†These authors contributed equally.

 

# Navigating cross-cultural research: methodological and ethical considerations

Tanya Broesch[1,†], Alyssa N. Crittenden[2,†], Bret A. Beheim[3],
Aaron D. Blackwell[4], John A. Bunce[3], Heidi Colleran[3,5], Kristin Hagel[3],
Michelle Kline[6], Richard McElreath[3], Robin G. Nelson[7], Anne C. Pisor[4],
Sean Prall[8], Ilaria Pretelli[3], Benjamin Purzycki[9], Elizabeth A. Quinn[10],
Cody Ross[3], Brooke Scelza[11], Kathrine Starkweather[3,12], Jonathan Stieglitz[13]
and Monique Borgerhoff Mulder[3,14]

[1]Department of Psychology, Simon Fraser University, BC, Canada
[2]Department of Anthropology, University of Nevada, Las Vegas, NV, USA
[3]Department of Human Behavior, Ecology and Culture, Max-Planck-Institute for Evolutionary Anthropology,
Leipzig, Germany
[4]Department of Anthropology, Washington State University, Pullman, WA, USA
[5]BirthRites Independent Max Planck Research Group, Max-Planck-Institute for Evolutionary Anthropology,
Leipzig, Germany
[6]Centre for Culture and Evolution, Brunel University, London, UK
[7]Department of Anthropology, Santa Clara University, CA, USA
[8]Department of Anthropology, University of Missouri, MO, USA
[9]Department of the Study of Religion, Aarhus University, Aarhus, Denmark
[10]Department of Anthropology, Washington University, Saint Louis, MO, USA
[11]Department of Anthropology, UCLA, Los Angeles, CA, USA
[12]Department of Anthropology, University of Illinois, Chicago, USA
[13]Institute for Advanced Study, Toulouse, France
[14]Department of Anthropology, University of California, Davis, CA, USA

BAB, 0000-0003-4653-3155; ADB, 0000-0002-5871-9865; JAB, 0000-0003-4092-485X;
HC, 0000-0002-2126-8116; SP, 0000-0001-5719-6460; BP, 0000-0002-9595-7360;
CR, 0000-0002-0067-4799; KS, 0000-0002-1554-4567; JS, 0000-0001-5985-9643;
MBM, 0000-0003-1117-5984

The intensifying pace of research based on cross-cultural studies in the social sciences necessitates a discussion of the unique challenges of multi-sited research. Given an increasing demand for social scientists to expand their data collection beyond WEIRD (Western, educated, industrialized, rich and democratic) populations, there is an urgent need for transdisciplinary conversations on the logistical, scientific and ethical considerations inherent to this type of scholarship. As a group of social scientists engaged in cross-cultural research in psychology and anthropology, we hope to guide prospective cross-cultural researchers through some of the complex scientific and ethical challenges involved in such work: (a) study site selection, (b) community involvement and (c) culturally appropriate research methods. We aim to shed light on some of the difficult ethical quandaries of this type of research. Our recommendation emphasizes a community-centred approach, in which the desires of the community regarding research approach and methodology, community involvement, results communication and distribution, and data sharing are held in the highest regard by the researchers. We argue that such considerations are central to scientific rigour and the foundation of the study of human behaviour.

## 1. Introduction

The acknowledgement that most research in psychology and other adjacent fields is overwhelmingly based on so-called WEIRD (Western, educated,

industrialized, rich and democratic) populations [1] has given way to intensified research funding, publication and visibility of collaborative cross-cultural studies across the social sciences that expand the geographical range of study populations (e.g. [2–10]). The rapid expansion of cross-cultural team science has been precipitated by the ever-increasing availability of online global data sources and the expansion of the cross-cultural enterprise into fields such as economics [11], political science [12] and other disciplines with little previous field research expectations or ethnographic focus. This expansion necessarily generates concerns regarding responsible methods and practice. For example, many of the so-called non-WEIRD communities who participate in research are Indigenous, from low- and middle-income countries in the global South, live in post-colonial contexts, and/or are marginalized within their political systems, creating power differentials between researchers and researched [13,14]. This creates a need for transdisciplinary discussion on the importance of community participation and the explanation and sharing of research outputs with participants.

Given increasing pressure for social scientists to expand the range of societies from which they recruit participants to test hypotheses about human behaviour, we convened a working group to discuss some of the unique scientific and ethical challenges of cross-cultural research. As a group of investigators engaged in such research in psychology and anthropology, our research objectives include testing theoretically derived hypotheses to examine general patterning and explain cultural variation in human behaviour. As such, we face challenges in how to collect systematic data, either as the primary fieldworker or in collaboration with shorter-term visitors who wish to collect their own data. The growing appetite for including diverse populations in work on demography, health, wealth, cooperation, cognition, infant and child development, and belief systems raises unique scientific and ethical issues, independent of discipline or research topic.

This paper adds to the growing dialogue on best practices when working with populations or cultural groups in low- to middle-income regions (see [13–18]) and touches on topics that many social scientists, particularly cultural anthropologists, have been writing about for decades [19,20]. Much cross-cultural research has historically been rooted in racist, capitalist ideas and motivations [19]. Scholars have long debated whether research aiming to standardize cross-cultural measurements and analysis is tacitly engaged and/or continues to be rooted in colonial and imperialist practices [21,22]. Given this history, it is critical that participating scientists reflect upon these issues and be accountable to their participants and colleagues for their research practices. We argue that cross-cultural research be grounded in the recognition of the historical, political, sociological and cultural forces acting on the communities and individuals of focus. These perspectives are often contrasted with 'science'; here we argue that they are necessary as a foundation for the study of human behaviour.

Here, we present considerations that we have found to be useful in our own work. More specifically, we propose that careful scrutiny of (a) study site selection, (b) community involvement and (c) culturally appropriate research methods will begin to address some of the complex scientific and ethical challenges of cross-cultural research. Particularly for those initiating collaborative cross-cultural projects, we focus here on pragmatic and implementable steps. We stress that our goal is not to review the literature on colonial or neo-colonial

research practices, to provide a comprehensive primer on decolonizing approaches to field research, nor to identify or admonish past misdemeanours in these respects—misdemeanours to which many of the authors of this piece would readily admit. Furthermore, we acknowledge that we ourselves are writing from a place of privilege as researchers educated and trained in disciplines with colonial pasts. Our goal is simply to help researchers in the future better plan and execute their projects with appropriate consideration and inclusion of study communities and culturally appropriate methodologies.

## (a) Study site selection

Study site selection in cross-cultural research involves three major conceptual issues. First, the increased interest in data collected from so-called non-Western societies means that study communities outside of WEIRD contexts are prized as sites for testing theories about human behaviour. This has sometimes led to an inclusion of 'non-WEIRD' populations in cross-cultural research without further regard for why specific populations should be included [23]. The binning of non-Western populations as a comparative sample to the cultural West (i.e. the 'West versus rest' approach) is often unwittingly reinforced by researchers who heeded the call to expand study site selection beyond WEIRD societies [1]. Here, we propose that researchers identify a clear *theoretical justification* for inclusion of any study population—WEIRD or not—based on knowledge of the relevant cultural and/or environmental context (see [24] for a good example). Regardless of whether a research group is investigating human universals or cultural variation, including any population in a study sample without justification of their inclusion is tantamount to binning and is, therefore, theoretically problematic [21].

Second, contemporary 'small-scale' communities continue to be discussed in the literature as proxies of our ancestral past—to varying degrees, often based on their food economy and the degree to which it is considered to be 'traditional' (e.g. foraging, small-scale horticulture). While some of these groups may occupy areas that are ecologically similar to the environments in which early modern humans lived and have social systems that may inform our understanding of those lifeways, these communities differ from early human communities in key ways. Many communities engage in mixed-subsistence practices [25] and currently reside in marginal environments that *may* not reflect their ancestral homelands [26]. Far from the romantic notion that such populations are uncontacted and living in harmony with the natural environment, in reality, they are impacted by ecological, social and political changes from outside/globalizing forces [27]. Studying contemporary communities as referential models of ancestral lifeways not only acts to further marginalize these societies, but can also lead to erroneous scientific conclusions—for example, about ancestral patterns of diet or cooperation (see [28–31]).

Third, when researchers design their cross-cultural studies, it is important to be cognizant that they are (to some extent) constrained by the relatively limited number of active field sites that can generate appropriate data. As such, cross-cultural investigators are working with a potentially biased sample of global populations from which broad inferences about humanity must be cautiously drawn (see [23]). This concern parallels our call for theoretical justification of the selection of samples; it

is both the diversity of samples and the *match between theory and cultural context* that make for improved research design (see [23] for full discussion and examples).

To address these three conceptual issues, we suggest that researchers and reviewers problematize the exoticizing of particular peoples and cultures [32]. Taking such an approach also works to minimize the inclusion of particular populations based on how popular or iconic they may be to researchers. One way to do this is to take a theoretically motivated approach to sampling communities. For example, one might select communities that vary along the specific axis of theoretical interest, such as age structure, female-biased kinship or extent of market integration (see [23]).

Intra-population sampling decisions are also important as they involve unique ethical and social challenges. For example, foreign researchers (as sources of power, information and resources) represent both opportunities for and threats to community members. These relationships are often complicated by power differentials due to unequal access to wealth, education and historical legacies of colonization [15–20]. As such, it is important that investigators are alert to the possible bias among individuals who initially interact with researchers, to the potential negative consequences for those excluded, and to the (often unspoken) power dynamics between the researcher and their study participants (as well as among and between study participants) [32–35].

We suggest that a necessary first step is to carefully consult existing resources outlining best practices for ethical principles of research. Many of these resources have been developed over years of dialogue in various academic and professional societies (e.g. American Anthropological Association, International Association for Cross Cultural Psychology, International Union of Psychological Science). Furthermore, communities themselves are developing and launching research-based codes of ethics [36,37] and providing carefully curated open-access materials (e.g. https://www.itk.ca), often written in consultation with ethicists in low- to middle-income countries (see [38]).

### (b) Community involvement

Too often researchers engage in 'extractive' research, whereby a researcher selects a study community and collects the necessary data to exclusively further their own scientific and/or professional goals without benefiting the community. This reflects a long history of colonialism in social science [15–20,33–35]. Extractive methods may not only lead to methodological challenges but also act to alienate participants from the scientific process and are often unethical. Many researchers are associated with institutions tainted with colonial, racist and sexist histories, sentiments and in some instances perpetuating into the present. Much cross-cultural research is carried out in former or contemporary colonies, and in the colonial language. Explicit and implicit power differentials create ethical challenges that can be acknowledged by researchers and in the design of their study (see [39] for an example in which the power and politics of various roles played by researchers is discussed). To provide examples of how to do this, we draw on frameworks from cultural anthropology and development studies, including participatory research, community collaboration and grounded theory [40–43]. What these frameworks hold in common, and what we reiterate here, is that it is critical

that communities be included in study design, implementation and presentation of research/return of results. There is no one-size-fits-all approach, yet a productive baseline may be for researchers to consider community inclusion as part of their project design from the start. Ideally, the community is not only central to the planned research, but is leading it. We realize that not all research approaches can include a research team that spans the research institution, the investigators and the community; however, we would like to note that in many instances, community-based participatory research is shifting towards this type of relationship between researchers and study communities [44,45].

Even if a research project does not include co-investigators from the study community, or establishing a long-term community collaboration is not an aim, the inclusion of research participants at the outset is possible. For example, in a population genetic study on the early population history of Vanuatu [46], one of the authors (H.C.) explored different approaches to explain the initial purpose of the research project before data collection. At a broad level, an analogy with linguistic family trees was most salient for discussion of population history and emerged naturally from conversations with communities about whether to carry out the research in the first place. Learning to describe the DNA itself in Indigenous idioms was far more challenging and was only possible by including the community in all stages of the project. Another co-author (A.N.C.), provided feedback on temporal changes in food and water insecurity in a foraging population in Tanzania using a different strategy: she enlisted community members as data collectors, whose feedback on interview questions was incorporated prior to data collection in order to ensure that the concepts being queried were understood by participants [47].

Context-specific knowledge is important when planning how to obtain and document informed consent in an ethical and culturally appropriate way. Most informed consent procedures were developed within the medical research community, with strict criteria for inclusion and high standards of linguistic comprehension expected. For people whose only experience of signing a formal agreement is from legal, political or medical contexts, standard consent forms can have unintended significance. Accordingly, researchers may consider an active community-level discussion as part of the consent process prior to the seeking of individual-level consent (see [48] for a full discussion). Consent is also often thought to be a one-time transaction, usually at the beginning of a study, experiment or interview. However, this is not an appropriate fit for communities where formal legal obligations carry less currency than do reciprocal social relationships. Consent should, therefore, be seen as a process and a dialogue, also referred to as 'dynamic consent', not merely the collection of names and signatures [49–52].

A new suite of challenges emerges once data collection has ended. There are ethical issues regarding the return of research results and associated data to the community. It is important that researchers discuss this with participants as part of the consent process and respect the desires of the community in this regard. It is often considered best practice for researchers to provide ample time for participants to query and discuss results, either or both in collaborative discussions with the community or private discussions with interested respondents [36–38,48]. Ideally, such community discussions provide the researcher with novel insights into data interpretation while providing participants with a satisfactory understanding of

the knowledge generated by the research and an opportunity to engage with the researchers' study motivations.

We also suggest that researchers consider how communities might benefit from access to the data they provide, and how local capacity to use such data can identified as part of the research [44,45]. Ultimately, we suggest a participant-led rather than top-down approach in making these decisions. By having conversations with participating communities about how they would like data returned, researchers and participants may find solutions for data sharing that are meaningful to communities—often through the production of archival works. For example, co-author A.C.P. collected video footage that was returned to the community; in a project on the production of handicrafts, the resultant video footage was uploaded to the internet, where community members indicated that they (and future generations) would have better access to the footage. Researchers and communities may consider uploading digital media to community-run websites or even to YouTube. When considering data sharing, however, it is important to note that some types of data-storage facilities (e.g. computers, libraries, YouTube) may not be accessible or appropriate to their participants. One strategy used independently by three of the authors (H.C., J.A.B. and A.N.C.) is to provide SD cards to participants with project-related video, photo and audio data which can be read by mobile phones. This allows information to be either kept secret by phone owners or to be shared. Another option used by co-author M.B.M. was to draw on her research to facilitate workshops for the writing and publication of a collectively sourced cultural history; she made copies of the book freely available to local schools [53]. A two-way dialogue between researchers and participants is needed to arrive at a reasonable solution based on participants' preferences.

Data sharing may also include shifting ownership of research outputs to participants in a more explicit manner. For example, there is a set of recommended practices for research conducted within Indigenous communities in Canada which stipulate that data remains the property of the participating communities [54]. It is important to meet the ethical standards of communities as well as those of government and research institutions (e.g. universities). For some types of data (e.g. open access data sharing), this may include carefully anonymizing results before transferring ownership in order to protect individual or community identities. However, we recognize that researchers will need to consider the ethics of publishing information from study communities alongside the requirements of funding agencies and institutional review boards, as well as the priorities of open science. We suggest that the research be designed (and budgeted) to allow time to return to the study communities to present and discuss the results and these issues, if possible, prior to publication. For example, the Wenner-Gren foundation has a grant designed to enable grantees to return to their research location (e.g. http://www.wennergren.org/programs/engaged-anthropology-grant).

Far too often, little attention is paid to the politics of representation when disseminating research results more widely, especially in online forums (including social media). It is important that all stakeholders, including all collaborating researchers, assume responsibility for the language used to describe results, whether by press offices or journalists or by the researchers themselves, as well as for the use of photographs, videos, audio recordings, material culture and artefacts in research and public outreach efforts. The recording and use of these materials should be addressed in the process of informed consent (see above). Sensationalizing or exoticizing images or language not only demeans study communities but can also undo years of careful community-based work. These practices are unethical because they may misrepresent participants; they can also affect relationships between study communities and field researchers. All researchers can bear these issues in mind and exert more control over public dissemination of their work. One suggestion to address these potential issues is for investigators themselves to write the press releases or, minimally, to review and approve press releases and associated images prepared by third parties.

## (c) Research design and methods

Data collection methods largely stemming from WEIRD intellectual traditions are being exported to a range of cultural contexts. This is often done with insufficient consideration of the translatability (e.g. equivalence or applicability) or implementation of such concepts and methods in different contexts, as already well documented [15–20]. It is critical that researchers translate the language, technological references and stimuli as well as examine the underlying cultural context of the original method for assumptions that rely upon WEIRD epistemologies [55,56]. This extends to non-complex visual aids, attempting to ensure that even scales measure what the researcher is intending (see [57] for discussion on the use of a popular economic experiment in small-scale societies).

For example, in a developmental psychology study conducted by Broesch and colleagues [58], the research team exported a task to examine the development and variability of self-recognition in children across cultures. Typically, this milestone is measured by surreptitiously placing a mark on a child's forehead and allowing them to discover their reflective image and the mark in a mirror. While self-recognition in WEIRD contexts typically manifests in children by 18 months of age, the authors tested found that only 2 out of 82 children (aged 1–6 years) 'passed' the test by removing the mark using the reflected image. Note that they began testing younger children and moved up the developmental trajectory, eventually testing older children who also did not 'pass the test' by Western standards. Their results are unexplained by existing developmental theories. The authors' interpretation of these results is that performance reflects false negatives and instead measures implicit compliance to the local authority figure who placed the mark on the child. This raises the possibility that the mirror test may lack construct validity in cross-cultural contexts—in other words, that it may not measure what it was designed to measure.

An understanding of cultural norms may ensure that experimental protocols and interview questions are culturally and linguistically salient. This can be achieved by implementing several complementary strategies. A first step may be to collaborate with members of the study community to check the relevance of the instruments being used. Incorporating perspectives from the study community from the outset can reduce the likelihood of making scientific errors in measurement and inference [54].

An additional approach is to use mixed methods in data collection, such that each method 'checks' the data collected using the other methods. A recent paper (see [59]) provides suggestions for a rigorous methodological approach to

conducting cross-cultural comparative psychology, underscoring the importance of using multiple methods with an eye towards a convergence of evidence. A mixed-method approach can incorporate a variety of methods such as participant observation, semi-structured interviews and experiments. For example, in their study on mate choice among Himba pastoralists of Namibia, Scelza and Prall [60] first employed semi-structured discussion groups and informal conversations with study participants. After better understanding the ways in which Himba themselves express desired characteristics of formal and informal partners, the researchers incorporated these characteristics into a ranking task [61]. Similarly, in a study of contraceptive use in rural Poland [62], qualitative interviews prior to formal data collection allowed the researchers to understand that the distinction between 'modern' and 'traditional' methods elicited very different (and apparently underreported) use than when the distinction was made between 'natural' and 'artificial'.

More generally, asking participants to talk aloud [63] as they complete a task or asking follow-up (debriefing) questions at the end of the experiment may allow researchers to better understand the decision-making processes at play (see [64,65] for recommendations and examples). Some guidelines for incorporating participant observation and qualitative interviews are available from Bernard [63] and Matsumoto & Van de Vijver [66]. For definitions, examples, and a full discussion of different kinds of bias in social science measures, see Van de Vijver & Tanzer [67]. There are also a number of Indigenous research methodologies that have been well-developed and extensively applied. For example, the Pagtatanong-tanong interview method developed and documented in the Philippines maximizes respect and equality by allowing equal time for participants and interviewers to engage in questioning (see [68]). We recommend using these resources as a guide *prior to* developing study methods and prioritizing the collection of baseline data, field testing instruments, and soliciting and incorporating community feedback before data collection commences.

## 2. Conclusion

Our aim here is to add to the growing dialogue on best practices in social science research, particularly as they relate to cross-cultural studies involving research participants from widely variable communities around the world. As research funding and publication of cross-cultural studies continues to expand across the social sciences, it is necessary to acknowledge the unique methodological and ethical challenges of this research. With scholars from a wide range of disciplines increasingly engaging in such research, often with little or no formal field training or experience working outside of post-industrialized contexts from the global North, special consideration of (a)

study site selection, (b) community involvement and (c) locally appropriate implementation of research design and methods is essential. Our intention is not to discourage researchers from embarking on cross-cultural studies, but rather to alert them to the multi-dimensional considerations at play, ranging from study design to participant inclusion, and to encourage constructive exchange and collaboration with participant communities. We suggest one solution may be for researchers new to cross-cultural studies to collaborate with field researchers who have established, long-term relationships with communities. We are not proposing that long-term researchers should be considered gatekeepers to the communities where they work—that role should only be played by the communities themselves. Rather, we are suggesting that individuals with established ties to a community may be useful guides for locally relevant materials, locally appropriate ethical and practical guidelines, and local contacts.

Transdisciplinary dialogue on principles and practices are useful not only for researchers (at all career stages) but also for funding agencies and reviewers evaluating twenty-first-century cross-cultural research across multiple domains of science. In short, deeper consideration of how to select sites for comparative investigations, how to engage target communities, and how to design research protocols in culturally sensitive ways will allow researchers to address some of the ethical and logistical challenges highlighted here—issues that all of the co-authors of this piece continue to grapple with in our own research and the communities with whom we work.

Data accessibility. This article has no additional data.

Authors' contributions. All authors contributed to the idea, outline and structure of the manuscript at the MPI workshop. T.B. and A.N.C. wrote the first draft of the manuscript with edits by M.B.M. The following authors provided comments and edits on manuscript drafts: J.A.B., H.C., K.H., M.K., R.G.N., A.C.P., C.R. and B.S. The following authors contributed to discussions at the workshop: B.A.B., A.D.B., R.M., S.P., I.P., B.P., E.A.Q., K.S., J.S. All authors edited and approved the final manuscript.

Funding. The workshop that generated the basis for this manuscript was funded by the Department of Human Behaviour, Ecology and Culture at the Max Planck Institute for Evolutionary Anthropology in Leipzig, Germany (proposal written by authors M.K. and R.G.N. and coordinated by M.K., R.G.N., K.S.). J.S. acknowledges funding from the French National Research Agency under the Investments for the Future (Investissements d'Avenir) programme (ANR-17-EURE-0010).

Competing interests. We declare we have no competing interests.

Acknowledgements. We thank the host communities with whom we have worked for their patience, collaboration and the knowledge that they have shared. We also thank Claudia Jacobi and the staff at MPI-EVA in Leipzig for their work in hosting the workshop, and Shani Msafiri Mangola, Elspeth Ready, Tim Caro and Daniel Benyshek for helpful feedback on earlier drafts of this manuscript. T.B. also thanks the Coady International Institute, particularly Allison Mathie and Gord Cunningham for hosting, teaching and supporting her transition to participant-engaged research.

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
