## [Reviewer comments · Proceedings of the Royal Society B: Biological Sciences]

Review History

RSPB-2020-1245.R0 (Original submission)

Review form: Reviewer 1

Recommendation

Major revision is needed (please make suggestions in comments)

Scientific importance: Is the manuscript an original and important contribution to its field?

Excellent

General interest: Is the paper of sufficient general interest?

Good

Quality of the paper: Is the overall quality of the paper suitable?

Excellent

Is the length of the paper justified?

Yes

Should the paper be seen by a specialist statistical reviewer?

No

Do you have any concerns about statistical analyses in this paper? If so, please specify them explicitly in your report.

No

It is a condition of publication that authors make their supporting data, code and materials available - either as supplementary material or hosted in an external repository. Please rate, if applicable, the supporting data on the following criteria.

Is it accessible?

N/A

Is it clear?

N/A

Is it adequate?

N/A

Do you have any ethical concerns with this paper?

No

Comments to the Author

The authors set out a template for ethical cross-cultural social science data collection. As attention continues to increase to the profound problems associated with WEIRD-centric research, a road map such as this is timely and valuable. On the whole, the piece is well-written and the arguments clearly set out. It will make a valuable contribution. There are some things I would like to see addressed though.

First, the general approach taken is one that implies cross-cultural research is an expedition in search of difference. If so motivated, the approaches advocated are valid. But what is missed are efforts at uncovering human universals: Those aspects of our behavior that we might attribute to a 'human' cognition, rather than a cultural or environmental one. In the case of the latter, having "clear theoretical reasons for inclusion for any study population that are based on knowledge of the relevant cultural context" is possibly overkill. Obviously, the theoretical reasons could be that if members of communities who contrast on a range of dimensions show similar patterns of response to test stimuli, autapomorphy can be evoked – but I don't think this is what the authors mean in the above statement. I feel that coverage of this issue (pursuit of human universals), and how it sits within the proposed arguments, is warranted.

Second, when writing a piece like this it is inevitable that a sense of caution finds its way in. Calling for change risks being undermined if certain individuals feel themselves singled-out and unfairly criticized. Nonetheless, missing from this work are explicit examples of where the authors feel published cases of cross-cultural research should be thought of as invalid or poorly executed. We are given examples of where the authors think they themselves have done a good job, but little of the opposite. This makes it hard to ascertain where the authors consider the boundary lies between good and bad cross-cultural work.

Third, there is a need to consider emerging researchers who are motivated to change the way of things but confronted with the overwhelming push to conform to high volume, quick turn-around publication strategies. Should such a researcher abandon willingness to undertake cross-cultural data collection if it doesn't adhere to the standards set out in this paper because it will simply take too much time and be less likely to yield outcomes publishable in high-impact papers? Or can they take their motivation and explore testing in new communities, armed with the knowledge of what is ideal (as set out in this paper) but knowing that so long as they show awareness of the issues de-homogenising participant pools is a worthwhile endeavor and better than more of the same (in this sense, any cross-cultural data collection is better than no cross-cultural data collection). I think the authors would adopt the latter position, but if so perhaps this

could be clearly outlined in the manuscript. That is, something that gives encouragement and support to early career researchers rather than something which leaves them feeling as if the pursuit of cross-cultural samples is unattainable.

Review form: Reviewer 2 (Ronald Fischer)

Recommendation

Major revision is needed (please make suggestions in comments)

Scientific importance: Is the manuscript an original and important contribution to its field?

Good

General interest: Is the paper of sufficient general interest?

Good

Quality of the paper: Is the overall quality of the paper suitable?

Marginal

Is the length of the paper justified?

Yes

Should the paper be seen by a specialist statistical reviewer?

No

Do you have any concerns about statistical analyses in this paper? If so, please specify them explicitly in your report.

No

It is a condition of publication that authors make their supporting data, code and materials available - either as supplementary material or hosted in an external repository. Please rate, if applicable, the supporting data on the following criteria.

Is it accessible?

N/A

Is it clear?

N/A

Is it adequate?

N/A

Do you have any ethical concerns with this paper?

No

Comments to the Author

I highly welcome this article for a broader audience, calling for more awareness of the methodological and ethical issues when dealing with non-WEIRD populations. My main concern is that it would be important to refer to existing discussions that have covered significant parts of the paper. It would be important to point interested researchers to guidelines and resources that have been carefully curated and are available for researchers to use. These issues have been discussed widely in relevant circles in psychological, sociological and anthropological contexts and it would be useful to make these resources more widely available.

To provide some key points, literatures and resources, here are some selected sources. These

sources are certainly not exhaustive, but may help in getting an overview of what resources and guidelines are available for the issues that are highlighted by the authors.

Ethical issues:

A number of international associations (including the International Association for Cross-Cultural Psychology, the International Union of Psychological Science and the International Association of Applied Psychology) have adopted a universal declaration of ethical principles for psychologists (see

<http://www.am.org/iupsys/resources/ethics/univdecl2008.html>), which is based on the United Nations declaration of human rights. This declaration has been adopted after a long process of consultation and discussions with both researchers and indigenous communities around the world. It is an important document and it would be highly important to draw greater attention to this declaration and encourage greater usage by novice researchers in the field.

Method issues:

The issue of standardization and equivalence of measures has received significant attention since the 1970s. Some of the discussion (e.g., the rural Poland example, trolley problem) suggest that the researchers did not properly think through the measurement development process, failing to identify functional equivalence issues and ignoring questions with structural invariance by jumping directly to using imposed etic measurement instruments. These issues have been discussed in great detail in the psychological and sociological literature and excellent sources are available that can guide researchers through this process.

Good general overviews of the issues for test development and test adaptation can be found here:

Fischer, R. & Poortinga, Y.H. (2018). Addressing methodological challenges in culture-comparative research. *Journal of Cross-Cultural Psychology*, 49, 691-712.

Fischer, R. & Karl, J.A. (2019). A primer to (cross-cultural) multi-group invariance testing possibilities in R. *Frontiers in Psychology*. 10.3389/fpsyg.2019.01507

Fontaine, J. R. J. (2005). Equivalence. In K. Kempf-Leonard (Ed.) *Encyclopedia of Social Measurement*. Academic Press.

Harkness, J., Pennell, B. E. & Schoua-Glusberg, A. 2004. Survey questionnaire translation and assessment. *Methods for testing and evaluating survey questionnaires*: 453-473. Hoboken, NJ: John Wiley & Sons.

Van de Vijver, F. & Leung, K. (1997). *Methods and Data analysis of comparative research*. Thousand Oaks, CA: Sage.

Van de Vijver, F. & Tanzer, N. K. (1994). Bias and equivalence in cross-cultural assessment: an overview. *Revue Européenne de Psychologie Appliquée/European Review of Applied Psychology*, 54, 119-135.

The book by Matsumoto and Van de Vijver on *Cross-Cultural Research Methods* (CUP, 2011) also provides checklists and guidelines for some of the issues raised in this article.

The authors scratch a number of significant issues and provide nice examples of some issues. It misses a larger systematic approach to outline the specific issues and what researchers can do to think through those problems.

The paper by Fisher & Portinga is the lead article of a special issue that deals with these and other issues across a number of areas within psychology in greater detail. The opening paper also includes some guiding questions and steps to consider when setting up and planning studies across diverse cultural communities.

The issue of standardization of testing procedures is important. Abubakar (2008) has described one compromise for the testing of young children in Kenya. She describes how tests that require strict standardization are administered last and the test administrator starts off by playing football with the child and her siblings, which allows for observations of balance, control and other aspects of motor function. Once the child has developed some trust and feels comfortable with the assessor, more standardized instructions to test cognitive abilities are used.

The issue of power dynamics are crucial. Both in anthropology and sociology, these have received much attention. I agree that psychology has been much less attentive to these issues. Some possible useful discussions can be found here:

Breidenbach, J. & Nyiri, P. (2008). *Maxikulti. Der Kampf der Kulturen ist das Problem – zeigt die Wirtschaft uns die Loesung?* Frankfurt/Main: Campus.

Schuller, M. (2010). From Activist to Applied Anthropologist to Anthropologist? On the Politics of Collaboration. *Practicing Anthropology*, 32, 43-47.

The emphasis on participant methodologies is welcome. There are a number of indigenous approaches that have been well-developed and have been extensively applied, which are in line with what the authors are discussing.

For example:

Durie, M (1994). *Whaiora: Maori health development*. Auckland: Oxford University Press.

Pe-Pua, R. (1990). *Pagtatanung-tanong: A method for cross-cultural research*. In V. G. Enriquez (ed), *Indigenous Psychology: A Book of Readings*. Akademya Ng Sikoholhiyang Pilipino. 231-249. Quezon City.

Yet, these issues are not without their own problems. For an example of criticism of the *Pagtatanung-tanong* method:

Margallo, S. P. (1990). *The challenge of a scientific indigenous field research*. In V. G. Enriquez (ed), *Indigenous Psychology: A Book of Readings*. Akademya Ng Sikoholhiyang Pilipino. 256-265. Quezon City.

In summary, this is an important conceptual article but it misses to refer to available frameworks, guidelines and declarations that can be used by researchers in their work with non-WEIRD communities.

Review form: Reviewer 3

Recommendation

Major revision is needed (please make suggestions in comments)

Scientific importance: Is the manuscript an original and important contribution to its field?

Good

General interest: Is the paper of sufficient general interest?

Good

Quality of the paper: Is the overall quality of the paper suitable?

Good

Is the length of the paper justified?

Yes

Should the paper be seen by a specialist statistical reviewer?

No

Do you have any concerns about statistical analyses in this paper? If so, please specify them explicitly in your report.

No

It is a condition of publication that authors make their supporting data, code and materials available - either as supplementary material or hosted in an external repository. Please rate, if applicable, the supporting data on the following criteria.

Is it accessible?

N/A

Is it clear?

N/A

Is it adequate?

N/A

Do you have any ethical concerns with this paper?

No

Comments to the Author

This paper points to some important practical and ethical challenges that arise when working with diverse human populations. The discussion in the paper is timely given recent calls for social scientists to reach out to a broader range of human populations. I have two major suggestions that may increase the article's impact. I also have a few minor points.

First, the paper needs more examples to illustrate points:

- 1) What's a good example of choosing fieldsites based on theoretical rationale and a bad example? Ideally this would include real examples, not just hypotheticals
- 2) What's an example of how assuming foragers reflected ancestral conditions led to erroneous conclusions?
- 3) What are real examples of how unreflective sampling within a community led to bias?
- 4) Good, real examples in the methods and community involvement section.

Second, the prose is very bossy – lots of “musts” and “should” and “imperatives”. Readers will likely more swayed by an argument for how following certain practices will lead to specific positive outcomes or how certain practices will lead to specific negative outcomes.

Minor points:

- 1) It's not clear why the discussion is structured in this order: (1) study site selection, (2) culturally appropriate methods, (3) community involvement. If you look at the research process, it seems like community involvement or study site selection should come first and culturally appropriate methods later. For example community involvement would likely improve the framing of research questions (so they actually are relevant for local conditions) and also the development of methods.
- 2) “we eschew programmatic discourses on neocolonialistic social science practices”. I think this needs more clarification. What's an example of a neocolonialistic social science practice? How is neocolonialistic different from colonialistic? I'm not sure whether the jargon is needed here.
- 3) In abstract, “inherent in” instead of “inherent of”? “inherent in of”?

Decision letter (RSPB-2020-1245.R0)

14-Jul-2020

Dear Monique,

Your manuscript has now been peer reviewed and the reviewers' comments (not including confidential comments to the Editor) are included at the end of this email for your reference. I think the fact that all three referees provide extensive comments is clear evidence that they agree the topic is timely and important, and want to see you and your co-authors go the extra mile to

produce something even better. Therefore, I'd like you to rinvite you to evise your manuscript to take account of the referees' suggestions. I realise that length is still a constraint, so you cannot add as much material as one might in an ideal world, but I think a revision that would satisfy the referees, and like-minded readers, is definitely feasible and worthwhile.

We do not allow multiple rounds of revision so we urge you to make every effort to fully address all of the comments at this stage. If deemed necessary, your manuscript will be sent back to one or more of the original reviewers for assessment. If the original reviewers are not available we may invite new reviewers. Please note that I cannot guarantee eventual acceptance of your manuscript at this stage.

Research ethics:

Use of animals and field studies:

It is a condition of publication that you make available the data and research materials supporting the results in the article (<https://royalsociety.org/journals/authors/author-guidelines/#data>). Datasets should be deposited in an appropriate publicly available repository and details of the associated accession number, link or DOI to the datasets must be included in the Data Accessibility section of the article (<https://royalsociety.org/journals/ethics-policies/data-sharing-mining/>). Reference(s) to datasets should also be included in the reference list of the article with DOIs (where available).

Please submit a copy of your revised paper within three weeks. If we do not hear from you within this time your manuscript will be rejected. If you are unable to meet this deadline please let us know as soon as possible, as we may be able to grant a short extension.

Best wishes,
Innes

Prof. Innes Cuthill
Reviews Editor, Proceedings B
mailto: proceedingsb@royalsociety.org

Reviewer(s)' Comments to Author:

Referee: 1

Comments to the Author(s)

The authors set out a template for ethical cross-cultural social science data collection. As attention continues to increase to the profound problems associated with WEIRD-centric research, a road map such as this is timely and valuable. On the whole, the piece is well-written and the arguments clearly set out. It will make a valuable contribution. There are some things I would like to see addressed though.

First, the general approach taken is one that implies cross-cultural research is an expedition in search of difference. If so motivated, the approaches advocated are valid. But what is missed are efforts at uncovering human universals: Those aspects of our behavior that we might attribute to a 'human' cognition, rather than a cultural or environmental one. In the case of the latter, having "clear theoretical reasons for inclusion for any study population that are based on knowledge of the relevant cultural context" is possibly overkill. Obviously, the theoretical reasons could be that if members of communities who contrast on a range of dimensions show similar patterns of response to test stimuli, autapomorphy can be evoked – but I don't think this is what the authors

mean in the above statement. I feel that coverage of this issue (pursuit of human universals), and how it sits within the proposed arguments, is warranted.

Second, when writing a piece like this it is inevitable that a sense of caution finds its way in. Calling for change risks being undermined if certain individuals feel themselves singled-out and unfairly criticized. Nonetheless, missing from this work are explicit examples of where the authors feel published cases of cross-cultural research should be thought of as invalid or poorly executed. We are given examples of where the authors think they themselves have done a good job, but little of the opposite. This makes it hard to ascertain where the authors consider the boundary lies between good and bad cross-cultural work.

Third, there is a need to consider emerging researchers who are motivated to change the way of things but confronted with the overwhelming push to conform to high volume, quick turn-around publication strategies. Should such a researcher abandon willingness to undertake cross-cultural data collection if it doesn't adhere to the standards set out in this paper because it will simply take too much time and be less likely to yield outcomes publishable in high-impact papers? Or can they take their motivation and explore testing in new communities, armed with the knowledge of what is ideal (as set out in this paper) but knowing that so long as they show awareness of the issues de-homogenising participant pools is a worthwhile endeavor and better than more of the same (in this sense, any cross-cultural data collection is better than no cross-cultural data collection). I think the authors would adopt the latter position, but if so perhaps this could be clearly outlined in the manuscript. That is, something that gives encouragement and support to early career researchers rather than something which leaves them feeling as if the pursuit of cross-cultural samples is unattainable.

Referee: 2

Comments to the Author(s)

I highly welcome this article for a broader audience, calling for more awareness of the methodological and ethical issues when dealing with non-WEIRD populations. My main concern is that it would be important to refer to existing discussions that have covered significant parts of the paper. It would be important to point interested researchers to guidelines and resources that have been carefully curated and are available for researchers to use. These issues have been discussed widely in relevant circles in psychological, sociological and anthropological contexts and it would be useful to make these resources more widely available.

To provide some key points, literatures and resources, here are some selected sources. These sources are certainly not exhaustive, but may help in getting an overview of what resources and guidelines are available for the issues that are highlighted by the authors.

Ethical issues:

A number of international associations (including the International Association for Cross-Cultural Psychology, the International Union of Psychological Science and the International Association of Applied Psychology) have adopted a universal declaration of ethical principles for psychologists (see

<http://www.am.org/iupsys/resources/ethics/univdecl2008.html>), which is based on the United Nations declaration of human rights. This declaration has been adopted after a long process of consultation and discussions with both researchers and indigenous communities around the world. It is an important document and it would be highly important to draw greater attention to this declaration and encourage greater usage by novice researchers in the field.

Method issues:

The issue of standardization and equivalence of measures has received significant attention since the 1970s. Some of the discussion (e.g., the rural Poland example, trolley problem) suggest that the researchers did not properly think through the measurement development process, failing to identify functional equivalence issues and ignoring questions with structural invariance by jumping directly to using imposed etic measurement instruments. These issues have been

discussed in great detail in the psychological and sociological literature and excellent sources are available that can guide researchers through this process.

Good general overviews of the issues for test development and test adaptation can be found here:

Fischer, R. & Poortinga, Y.H. (2018). Addressing methodological challenges in culture-comparative research. *Journal of Cross-Cultural Psychology*, 49, 691-712.

Fischer, R. & Karl, J.A. (2019). A primer to (cross-cultural) multi-group invariance testing possibilities in R. *Frontiers in Psychology*. 10.3389/fpsyg.2019.01507

Fontaine, J. R. J. (2005). Equivalence. In K. Kempf-Leonard (Ed.) *Encyclopedia of Social Measurement*. Academic Press.

Harkness, J., Pennell, B. E. & Schoua-Glusberg, A. 2004. Survey questionnaire translation and assessment. *Methods for testing and evaluating survey questionnaires*: 453–473. Hoboken, NJ: John Wiley & Sons.

Van de Vijver, F. & Leung, K. (1997). *Methods and Data analysis of comparative research*. Thousand Oaks, CA: Sage.

Van de Vijver, F. & Tanzer, N. K. (1994). Bias and equivalence in cross-cultural assessment: an overview. *Revue Européenne de Psychologie Appliquée/European Review of Applied Psychology*, 54, 119-135.

The book by Matsumoto and Van de Vijver on *Cross-Cultural Research Methods* (CUP, 2011) also provides checklists and guidelines for some of the issues raised in this article.

The authors scratch a number of significant issues and provide nice examples of some issues. It misses a larger systematic approach to outline the specific issues and what researchers can do to think through those problems.

The paper by Fisher & Portinga is the lead article of a special issue that deals with these and other issues across a number of areas within psychology in greater detail. The opening paper also includes some guiding questions and steps to consider when setting up and planning studies across diverse cultural communities.

The issue of standardization of testing procedures is important. Abubakar (2008) has described one compromise for the testing of young children in Kenya. She describes how tests that require strict standardization are administered last and the test administrator starts off by playing football with the child and her siblings, which allows for observations of balance, control and other aspects of motor function. Once the child has developed some trust and feels comfortable with the assessor, more standardized instructions to test cognitive abilities are used.

The issue of power dynamics are crucial. Both in anthropology and sociology, these have received much attention. I agree that psychology has been much less attentive to these issues.

Some possible useful discussions can be found here:

Breidenbach, J. & Nyiri, P. (2008). *Maxikulti. Der Kampf der Kulturen ist das Problem - zeigt die Wirtschaft uns die Loesung?* Frankfurt/Main: Campus.

Schuller, M. (2010). From Activist to Applied Anthropologist to Anthropologist? On the Politics of Collaboration. *Practicing Anthropology*, 32, 43-47.

The emphasis on participant methodologies is welcome. There are a number of indigenous approaches that have been well-developed and have been extensively applied, which are in line with what the authors are discussing.

For example:

Durie, M (1994). *Whaiora: Maori health development*. Auckland: Oxford University Press.

Pe-Pua, R. (1990). *Pagtatanung-tanong: A method for cross-cultural research*. In V. G. Enriquez (ed), *Indigenous Psychology: A Book of Readings*. Akademya Ng Sikoholhiyang Pilipino. 231-249. Quezon City.

Yet, these issues are not without their own problems. For an example of criticism of the *Pagtatanung-tanong* method:

Margallo, S. P. (1990). The challenge of a scientific indigenous field research. In V. G. Enriquez (ed), *Indigenous Psychology: A Book of Readings*. Akademya Ng Sikoholhiyang Pilipino. 256-265. Quezon City.

In summary, this is an important conceptual article but it misses to refer to available frameworks, guidelines and declarations that can be used by researchers in their work with non-WEIRD communities.

Referee: 3

Comments to the Author(s)

This paper points to some important practical and ethical challenges that arise when working with diverse human populations. The discussion in the paper is timely given recent calls for social scientists to reach out to a broader range of human populations. I have two major suggestions that may increase the article's impact. I also have a few minor points.

First, the paper needs more examples to illustrate points:

- 1) What's a good example of choosing fieldsites based on theoretical rationale and a bad example? Ideally this would include real examples, not just hypotheticals
- 2) What's an example of how assuming foragers reflected ancestral conditions led to erroneous conclusions?
- 3) What are real examples of how unreflective sampling within a community led to bias?
- 4) Good, real examples in the methods and community involvement section.

Second, the prose is very bossy – lots of "musts" and "should" and "imperatives". Readers will likely more swayed by an argument for how following certain practices will lead to specific positive outcomes or how certain practices will lead to specific negative outcomes.

Minor points:

- 1) It's not clear why the discussion is structured in this order: (1) study site selection, (2) culturally appropriate methods, (3) community involvement. If you look at the research process, it seems like community involvement or study site selection should come first and culturally appropriate methods later. For example community involvement would likely improve the framing of research questions (so they actually are relevant for local conditions) and also the development of methods.
- 2) "we eschew programmatic discourses on neocolonialistic social science practices". I think this needs more clarification. What's an example of a neocolonialistic social science practice? How is neocolonialistic different from colonialistic? I'm not sure whether the jargon is needed here.
- 3) In abstract, "inherent in" instead of "inherent of"? "inherent in of"?

Author's Response to Decision Letter for (RSPB-2020-1245.R0)

See Appendix A.

RSPB-2020-1245.R1 (Revision)

Review form: Reviewer 2 (Ronald Fischer)

Recommendation

Accept with minor revision (please list in comments)

Scientific importance: Is the manuscript an original and important contribution to its field?

Good

General interest: Is the paper of sufficient general interest?

Good

Quality of the paper: Is the overall quality of the paper suitable?

Acceptable

Is the length of the paper justified?

Yes

Should the paper be seen by a specialist statistical reviewer?

No

Do you have any concerns about statistical analyses in this paper? If so, please specify them explicitly in your report.

No

It is a condition of publication that authors make their supporting data, code and materials available - either as supplementary material or hosted in an external repository. Please rate, if applicable, the supporting data on the following criteria.

Is it accessible?

N/A

Is it clear?

N/A

Is it adequate?

N/A

Do you have any ethical concerns with this paper?

No

Comments to the Author

Many thanks for revising your manuscript in light of the comments provided by the reviewers.

A few further comments:

You state that any sample (WEIRD or non-WEIRD) needs theoretical justification. I wholeheartedly agree with this. The immediately following statement: 'including non-WEIRD populations in a study sample without justifying the inclusion of each specific population is tantamount to binning and is, therefore, theoretically problematic' immediately seems to negate the prior statement and only focuses on non-WEIRD populations. I would recommend rephrasing this statement to make it clear that studying only WEIRD samples is equally problematic and normalizes the exclusion of a large majority of the world's population.

Please check the spelling of pagtatanong-tanong.

I hope that this paper will raise greater awareness of the multiple issues involved with cultural research.

Review form: Reviewer 3

Recommendation

Accept as is

Scientific importance: Is the manuscript an original and important contribution to its field?

Excellent

General interest: Is the paper of sufficient general interest?

Excellent

Quality of the paper: Is the overall quality of the paper suitable?

Excellent

Is the length of the paper justified?

Yes

Should the paper be seen by a specialist statistical reviewer?

No

Do you have any concerns about statistical analyses in this paper? If so, please specify them explicitly in your report.

No

It is a condition of publication that authors make their supporting data, code and materials available - either as supplementary material or hosted in an external repository. Please rate, if applicable, the supporting data on the following criteria.

Is it accessible?

N/A

Is it clear?

N/A

Is it adequate?

N/A

Do you have any ethical concerns with this paper?

No

Comments to the Author

The authors have thoughtfully addressed the concerns I raised in a prior review.

Decision letter (RSPB-2020-1245.R1)

27-Aug-2020

Dear Monique,

I am pleased to inform you that your manuscript RSPB-2020-1245.R1 entitled "Navigating cross-cultural research: methodological and ethical considerations" has been accepted for publication in Proceedings B.

The referees are happy with your revisions and have recommended publication, but referee 2 also suggests one final, very easy, modification which seems sensible. Therefore, I invite you to make that change and upload the final version of your manuscript. Because the schedule for publication is very tight, it is a condition of publication that you submit the revised version of your manuscript within 7 days. If you do not think you will be able to meet this date please let us know.

To upload your final manuscript, log into <https://mc.manuscriptcentral.com/prsb> and enter your Author Centre, where you will find your manuscript title listed under "Manuscripts with Decisions." Under "Actions," click on "Create a Revision." Your manuscript number has been appended to denote a revision. You will be unable to make your revisions on the originally submitted version of the manuscript. Instead, revise your manuscript and upload a new version through your Author Centre.

NB. From April 1 2013, peer reviewed articles based on research funded wholly or partly by RCUK must include, if applicable, a statement on how the underlying research materials – such

as data, samples or models – can be accessed. This statement should be included in the data accessibility section.

[http://datadryad.org/submit?journalID=RSPB&manu=\(Document not available\)](http://datadryad.org/submit?journalID=RSPB&manu=(Document+not+available)) which will take you to your unique entry in the Dryad repository. If you have already submitted your data to dryad you can make any necessary revisions to your dataset by following the above link. Please see <https://royalsociety.org/journals/ethics-policies/data-sharing-mining/> for more details.

Best wishes,
Innes

Prof. Innes Cuthill
Reviews Editor, Proceedings B
mailto: proceedingsb@royalsociety.org

Reviewer(s)' Comments to Author:

Referee: 3

Comments to the Author(s)

The authors have thoughtfully addressed the concerns I raised in a prior review.

Referee: 2

Comments to the Author(s)

Many thanks for revising your manuscript in light of the comments provided by the reviewers.

A few further comments:

You state that any sample (WEIRD or non-WEIRD) needs theoretical justification. I wholeheartedly agree with this. The immediately following statement: 'including non-WEIRD populations in a study sample without justifying the inclusion of each specific population is tantamount to binning and is, therefore, theoretically problematic' immediately seems to negate the prior statement and only focuses on non-WEIRD populations. I would recommend rephrasing this statement to make it clear that studying only WEIRD samples is equally problematic and normalizes the exclusion of a large majority of the world's population.

Please check the spelling of pagtatanong-tanong.

I hope that this paper will raise greater awareness of the multiple issues involved with cultural research.

Decision letter (RSPB-2020-1245.R2)

01-Sep-2020

Dear Dr Borgerhoff Mulder

I am pleased to inform you that your manuscript entitled "Navigating cross-cultural research: methodological and ethical considerations" has been accepted for publication in Proceedings B.

If you are likely to be away from e-mail contact during this period, let us know. Due to rapid publication and an extremely tight schedule, if comments are not received, we may publish the paper as it stands.

Your article has been estimated as being 7 pages long. Our Production Office will be able to confirm the exact length at proof stage.

Open access

You are invited to opt for open access via our author pays publishing model. Payment of open access fees will enable your article to be made freely available via the Royal Society website as soon as it is ready for publication. For more information about open access publishing please visit our website at http://royalsocietypublishing.org/site/authors/open_access.xhtml.

The open access fee is £1,700 per article (plus VAT for authors within the EU). If you wish to opt for open access then please let us know as soon as possible.

Paper charges

Sincerely,
Proceedings B
<mailto:proceedingsb@royalsociety.org>

Appendix A

August 5, 2020

Dear Editor Cuthill,

We found the comments and suggestions of the reviewers to be helpful and we feel that our manuscript is much improved. Below we address each concern and query, point by point. In general, we have adhered to the word limit by removing a few hypothetical examples and one actual example (i.e. Trolley car) in the methods sections and by removing direct reference to collaboration with long-term field researchers – given the suggestion that this appeared as the authors placing ourselves as gatekeepers. Furthermore, we have expanded the introduction and added several references throughout suggested by reviewers. All changes are indicated in red text in the track-changed document.

Thank you,

RESPONSE TO REVIEWERS

Referee: 1

Comments to the Author(s)

The authors set out a template for ethical cross-cultural social science data collection. As attention continues to increase to the profound problems associated with WEIRD-centric research, a road map such as this is timely and valuable. On the whole, the piece is well-written and the arguments clearly set out. It will make a valuable contribution. There are some things I would like to see addressed though.

First, the general approach taken is one that implies cross-cultural research is an expedition in search of difference. If so motivated, the approaches advocated are valid. But what is missed are efforts at uncovering human universals: Those aspects of our behavior that we might attribute to a 'human' cognition, rather than a cultural or environmental one. In the case of the latter, having “clear theoretical reasons for inclusion for any study population that are based on knowledge of the relevant cultural context” is possibly overkill. Obviously, the theoretical reasons could be that if members of communities who contrast on a range of dimensions show similar patterns of response to test stimuli, autapomorphy can be evoked – but I don't think this is what the authors mean in the above statement. I feel that coverage of this issue (pursuit of human universals), and how it sits within the proposed arguments, is warranted.

We now clarify our perspective regarding the inclusion of sample selection justification. We have expanded discussion to more clearly state our argument.

See Lines 106 - 128.

The relationships between studies of universality, variability and the universality of principles governing variability is an important issue, but not one intrinsic to the topic of

this paper. Accordingly, we simply clarify that the studies we address may be considering either, see line 106:

“Regardless of whether a research group is investigating human universals or cultural variation, including non-WEIRD populations in a study sample without justifying the inclusion of each specific population is tantamount to binning and is, therefore, theoretically problematic (18).”

Second, when writing a piece like this it is inevitable that a sense of caution finds its way in. Calling for change risks being undermined if certain individuals feel themselves singled-out and unfairly criticized. Nonetheless, missing from this work are explicit examples of where the authors feel published cases of cross-cultural research should be thought of as invalid or *poorly* executed. We are given examples of where the authors think they themselves have done a good job, but little of the opposite. This makes it hard to ascertain where the authors consider the boundary lies between good and bad cross-cultural work.

Thank you for this suggestion. Our aim is not to publicly admonish or demonize researchers, studies, or practices – which would act to alienate rather than educate. We now make this explicitly (line 81 - 93). Rather, our goal is to provide a roadmap for cross-cultural researchers moving forward.

Third, there is a need to consider emerging researchers who are motivated to change the way of things but confronted with the overwhelming push to conform to high volume, quick turn-around publication strategies. Should such a researcher abandon willingness to undertake cross-cultural data collection if it doesn't adhere to the standards set out in this paper because it will simply take too much time and be less likely to yield outcomes publishable in high-impact papers? Or can they take their motivation and explore testing in new communities, armed with the knowledge of what is ideal (as set out in this paper) but knowing that so long as they show awareness of the issues de-homogenising participant pools is a worthwhile endeavor and better than more of the same (in this sense, any cross-cultural data collection is better than no cross-cultural data collection). I think the authors would adopt the latter position, but if so perhaps this could be clearly outlined in the manuscript. That is, something that gives encouragement and support to early career researchers rather than something which leaves them feeling as if the pursuit of cross-cultural samples is unattainable.

Thank-you. We point the reviewer to our conclusion where we have incorporated this good suggestion. See lines 325 - 334:

“Our intention is not to discourage researchers from embarking on cross-cultural studies, but rather to alert them to the multi-dimensional considerations at play, ranging from study design to participant inclusion, and to encourage constructive exchange and collaboration with participant communities. We suggest one solution may be for researchers new to cross-cultural studies to collaborate with field researchers who have established, long-term relationships with communities. We are not proposing that long-term researchers should be considered gatekeepers to the communities where they work – that role should only be played by the communities themselves. Rather, we are suggesting that individuals with established ties to a community may be useful guides for

locally relevant materials, locally appropriate ethical and practical guidelines, and local contacts. “

Referee: 2

Comments to the Author(s)

I highly welcome this article for a broader audience, calling for more awareness of the methodological and ethical issues when dealing with non-WEIRD populations. My main concern is that it would be important to refer to existing discussions that have covered significant parts of the paper. It would be important to point interested researchers to guidelines and resources that have been carefully curated and are available for researchers to use. These issues have been discussed widely in relevant circles in psychological, sociological and anthropological contexts and it would be useful to make these resources more widely available.

To provide some key points, literatures and resources, here are some selected sources. These sources are certainly not exhaustive, but may help in getting an overview of what resources and guidelines are available for the issues that are highlighted by the authors.

Ethical issues:

A number of international associations (including the International Association for Cross-Cultural Psychology, the International Union of Psychological Science and the International Association of Applied Psychology) have adopted a universal declaration of ethical principles for psychologists (see <http://www.am.org/iupsys/resources/ethics/univdecl2008.html>), which is based on the United Nations declaration of human rights. This declaration has been adopted after a long process of consultation and discussions with both researchers and indigenous communities around the world. It is an important document and it would be highly important to draw greater attention to this declaration and encourage greater usage by novice researchers in the field.

We thank the reviewer for this important point, and now refer to a set of these declarations of ethical principles across a range of disciplines. We have also added additional citations of research codes of ethics written by indigenous communities and ethicists in low to middle income countries.

See Lines 145-152:

“We suggest that a necessary first step is to carefully consult existing resources outlining best practices for ethical principles of research. Many of these resources have been developed over years of dialogue in various academic and professional societies (e.g. American Anthropological Association, International Association for Cross Cultural Psychology, International Union of Psychological Science). Furthermore, communities themselves are developing and launching research-based codes of ethics (33, 34) and providing carefully curated open-access materials (e.g. <https://www.itk.ca>), often written in consultation with ethicists in low to middle income countries (see 35).”

Method issues:

The issue of standardization and equivalence of measures has received significant attention since the 1970s. Some of the discussion (e.g., the rural Poland example, trolley problem) suggest that the researchers did not properly think through the measurement development process, failing to identify functional equivalence issues and ignoring questions with structural invariance by jumping directly to using imposed etic measurement instruments. These issues have been discussed in great detail in the psychological and sociological literature and excellent sources are available that can guide researchers through this process.

Yes, we are aware of these studies and see the remit of the current general Perspective paper to alert readers from multiple disciplines, including biology, to the issue. Accordingly, we have incorporated many of the citations the reviewer suggests (thank you), and have removed probably the best known and most egregious example – the trolley car case – focusing more on additional concrete examples from published works. See page 9.

Good general overviews of the issues for test development and test adaptation can be found here:

Fischer, R. & Poortinga, Y.H. (2018). Addressing methodological challenges in culture-comparative research. *Journal of Cross-Cultural Psychology*, 49, 691-712.

Fischer, R. & Karl, J.A. (2019). A primer to (cross-cultural) multi-group invariance testing possibilities in R. *Frontiers in Psychology*. 10.3389/fpsyg.2019.01507

Fontaine, J. R. J. (2005). Equivalence. In K. Kempf-Leonard (Ed.) *Encyclopedia of Social Measurement*. Academic Press.

Harkness, J., Pennell, B. E. & Schoua-Glusberg, A. 2004. Survey questionnaire translation and assessment. *Methods for testing and evaluating survey questionnaires: 453–473*. Hoboken, NJ: John Wiley & Sons.

Van de Vijver, F. & Leung, K. (1997). *Methods and Data analysis of comparative research*. Thousand Oaks, CA: Sage.

Van de Vijver, F. & Tanzer, N. K. (1994). Bias and equivalence in cross-cultural assessment: an overview. *Revue Européenne de Psychologie Appliquée/European Review of Applied Psychology*, 54, 119-135.

The book by Matsumoto and Van de Vijver on *Cross-Cultural Research Methods* (CUP, 2011) also provides checklists and guidelines for some of the issues raised in this article.

We have now pointed readers to these useful resources.

The authors scratch a number of significant issues and provide nice examples of some issues. It misses a larger systematic approach to outline the specific issues and what researchers can do to think through those problems. The paper by Fisher & Portinga is the lead article of a special issue that deals with these and other issues across a number of areas within psychology in greater detail.

Thank-you. We now cite Fischer & Poortinga pointing the reader to theoretically motivated and rigorous methodological approaches that researchers can adopt. See Lines 292-295:

“A recent paper (see 56) provides suggestions for a rigorous methodological approach to conducting cross-cultural comparative psychology, underscoring the importance of using multiple methods with an eye toward a convergence of evidence.”

The opening paper also includes some guiding questions and steps to consider when setting up and planning studies across diverse cultural communities. The issue of standardization of testing procedures is important. Abubakar (2008) has described one compromise for the testing of young children in Kenya. She describes how tests that require strict standardization are administered last and the test administrator starts off by playing football with the child and her siblings, which allows for observations of balance, control and other aspects of motor function. Once the child has developed some trust and feels comfortable with the assessor, more standardized instructions to test cognitive abilities are used.

Thank-you for the excellent example that underscores our perspective regarding the critical importance of participant involvement and observation. We feel that this is now addressed in our overview of study method development and incorporating participant observation and bias. See Lines 307-310:

“Some guidelines for incorporating participant observation and qualitative interviews are available from Bernard (60) and Matsumoto and Van de Vijver (63). For definitions, examples, and a full discussion of different kinds of bias in social science measures, see van de Vijver & Tanzer (64).”

The issue of power dynamics is crucial. Both in anthropology and sociology, these have received much attention. I agree that psychology has been much less attentive to these issues.

We agree and thank-you for the references below. We now refer the reader to Schuller (2010) for an excellent in-depth description of power and membership.

Furthermore, regarding “much attention” we have now specify our goals a little more precisely in the Introduction, to clarify that we cannot review the huge literature on this topic

See Lines 81-93:

“Here we present considerations that we have found to be useful in our own work. More specifically we propose that careful scrutiny of (a) study site selection, (b) community involvement, and (c) culturally appropriate research methods will begin to address some of the complex scientific and ethical challenges of cross-cultural research. Particularly for those initiating collaborative cross-cultural projects, we focus here on pragmatic and implementable steps. We stress that our goal is not to review the literature on colonial or neo-colonial research practices, to provide a comprehensive primer on decolonizing approaches to field research, nor to identify or admonish past misdemeanours in these respects – misdemeanours to which many of the authors of this piece would readily admit. Furthermore, we acknowledge that we ourselves are writing from a place of

privilege as researchers educated and trained in disciplines with colonial pasts. Our goal is simply to help researchers in the future better plan and execute their projects with appropriate consideration and inclusion of study communities and culturally appropriate methodologies.”

Some possible useful discussions can be found here:

Breidenbach, J. & Nyiri, P. (2008). Maxikulti. Der Kampf der Kulturen ist das Problem – zeigt die Wirtschaft uns die Loesung? Frankfurt/Main: Campus.

Schuller, M. (2010). From Activist to Applied Anthropologist to Anthropologist? On the Politics of Collaboration. *Practicing Anthropology*, 32, 43-47.

The emphasis on participant methodologies is welcome. There are a number of indigenous approaches that have been well-developed and have been extensively applied, which are in line with what the authors are discussing.

For example:

Durie, M (1994). *Whaiora: Maori health development*. Auckland: Oxford University Press.

Pe-Pua, R. (1990). Pagtatanung-tanong: A method for cross-cultural research. In V. G. Enriquez (ed), *Indigenous Psychology: A Book of Readings*. Akademya Ng Sikoholhiyang Pilipino. 231-249. Quezon City.

Yet, these issues are not without their own problems. For an example of criticism of the Pagtatanung-tanong method:

Margallo, S. P. (1990). The challenge of a scientific indigenous field research. In V. G. Enriquez (ed), *Indigenous Psychology: A Book of Readings*. Akademya Ng Sikoholhiyang Pilipino. 256-265. Quezon City.

In summary, this is an important conceptual article but it misses to refer to available frameworks, guidelines and declarations that can be used by researchers in their work with non-WEIRD communities.

Thank you for the suggested citations. We have cited several of them where appropriate and now point directly to indigenous approaches.

Referee: 3

Comments to the Author(s)

This paper points to some important practical and ethical challenges that arise when working with diverse human populations. The discussion in the paper is timely given recent calls for social scientists to reach out to a broader range of human populations. I have two major suggestions that may increase the article’s impact. I also have a few minor points.

First, the paper needs more examples to illustrate points:

1) What's a good example of choosing fieldsites based on theoretical rationale and a bad example? Ideally this would include real examples, not just hypotheticals.

We now refer the reader to Barrett, 2020 as the author provides examples and rationale supporting theoretically motivated field research throughout

See Lines 126-129:

“This concern parallels our call for theoretical justification of the selection of samples; it is both the diversity of samples and the *match between theory and cultural context* that make for improved research design (See 20 for full discussion and examples).”

2) What's an example of how assuming foragers reflected ancestral conditions led to erroneous conclusions?

We agree that this needs to be more clear. Rather than point out other researchers who have failed, we have now included two examples of how assuming ancestral conditions can lead to erroneous conclusions and have also provided more explicit language on how such classification can be unethical for the participating communities. The following text and associated supporting citations were added.

Lines 109-119:

“Second, contemporary “small-scale” communities continue to be discussed in the literature as proxies of our ancestral past - to varying degrees, often based on their food economy and the degree to which it is considered to be “traditional” (e.g. foraging, small-scale horticulture). While some of these groups may occupy areas that are ecologically similar to the environments in which early modern humans lived and have social systems that may inform our understanding of those lifeways, these communities differ from early human communities in key ways. Many communities engage in mixed-subsistence practices (22) and currently reside in marginal environments that *may* not reflect their ancestral homelands (23). Far from the romantic notion that such populations are uncontacted and living in harmony with the natural environment, in reality, they are impacted by ecological, social, and political changes from outside/globalizing forces (24).”

3) What are real examples of how unreflective sampling within a community led to bias?

We have now included examples. See lines 119-121:

“Studying contemporary communities as referential models of ancestral lifeways not only acts to further marginalize these societies, but can also lead to erroneous scientific conclusions -- for example, about ancestral patterns of diet or cooperation (see 25-28).”

4) Good, real examples in the methods and community involvement section.

We now frame our paper as, “considerations” rather than as a template or guideline for conducting cross-cultural research. We have now pointed readers to a long list of existing

“how to” guides, open access materials, and works by indigenous scholars that include ethical codes of research conduct. See Lines 312-318:

“There are also a number of indigenous research methodologies that have been well-developed and extensively applied. For example, the PaTanong-Tanong interview method developed and documented in the Philippines maximizes respect and equality by allowing equal time for participants and interviewers to engage in questioning (see 65). We recommend using these resources as a guide *prior to* developing study methods and prioritizing the collection of baseline data, field testing instruments, and soliciting and incorporating community feedback before data collection commences.”

Second, the prose is very bossy—lots of “musts” and “should” and “imperatives”. Readers will likely more swayed by an argument for how following certain practices will lead to specific positive outcomes or how certain practices will lead to specific negative outcomes.

We thank the reviewer for this helpful point; it is something that has concerned us throughout preparation of the MS. We have worked to eradicate this messaging throughout the manuscript. Furthermore, we have included a sentence in the introduction (Lines 86-89) that indicates that we have learned from our many mistakes, and in the final paragraph that we ourselves are still learning (Lines 345-347). We hope that this mitigates any apparent “bossiness” that existed in the earlier draft of the manuscript.

Minor points:

1) It’s not clear why the discussion is structured in this order: (1) study site selection, (2) culturally appropriate methods, (3) community involvement. If you look at the research process, it seems like community involvement or study site selection should come first and culturally appropriate methods later. For example community involvement would likely improve the framing of research questions (so they actually are relevant for local conditions) and also the development of methods.

Thank-you for this suggestion. We agree and have restructured our manuscript to reflect this logical flow of our argument.

2) “we eschew programmatic discourses on neocolonialistic social science practices “. I think this needs more clarification. What’s an example of a neocolonialistic social science practice? How is neocolonialistic different from colonialistic? I’m not sure whether the jargon is needed here.

We have re-written this sentence so that jargon is removed. We hope that our point is now more clear. The paper now states our objectives more precisely in the introductory section:

Lines 81-93:

“Here we present considerations that we have found to be useful in our own work. More specifically we propose that careful scrutiny of (a) study site selection, (b) community involvement, and (c) culturally appropriate research methods will begin to address some of the complex scientific and ethical challenges of cross-cultural research. Particularly for

those initiating collaborative cross-cultural projects, we focus here on pragmatic and implementable steps. We stress that our goal is not to review the literature on colonial or neo-colonial research practices, to provide a comprehensive primer on decolonizing approaches to field research, nor to identify or admonish past misdemeanours in these respects – misdemeanours to which many of the authors of this piece would readily admit. Furthermore, we acknowledge that we ourselves are writing from a place of privilege as researchers educated and trained in disciplines with colonial pasts. Our goal is simply to help researchers in the future better plan and execute their projects with appropriate consideration and inclusion of study communities and culturally appropriate methodologies.”

3) In abstract, “inherent in” instead of “inherent of”? “inherent in

Fixed. Thank you.